# Associations of Dietary Intake with Urinary Melamine and Derivative Concentrations among Children in the GAPPS Cohort

**DOI:** 10.3390/ijerph19094964

**Published:** 2022-04-19

**Authors:** Melissa M. Melough, Drew B. Day, Amanda M. Fretts, Sarah Wang, Joseph T. Flynn, Ian H. de Boer, Hongkai Zhu, Kurunthachalam Kannan, Sheela Sathyanarayana

**Affiliations:** 1Department of Child Health, Behavior and Development, Seattle Children’s Research Institute, Seattle, WA 98101, USA; drew.day@seattlechildrens.org (D.B.D.); sarah.wang@seattlechildrens.org (S.W.); sheela.sathyanarayana@seattlechildrens.org (S.S.); 2Department of Epidemiology, University of Washington School of Public Health, Seattle, WA 98195, USA; amfretts@uw.edu; 3Department of Pediatrics, University of Washington, Seattle, WA 98195, USA; joseph.flynn@seattlechildrens.org; 4Division of Nephrology, Seattle Children’s Hospital, Seattle, WA 98105, USA; 5Division of Medicine, Kidney Research Institute, University of Washington, Seattle, WA 98105, USA; deboer@u.washington.edu; 6Department of Pediatrics, New York University School of Medicine, New York, NY 10016, USA; zhhkok@163.com (H.Z.); kurunthachalam.kannan@nyulangone.org (K.K.)

**Keywords:** melamine, ammelide, cyanuric acid, diet

## Abstract

Melamine is a nephrotoxic industrial chemical. Diet is one source of melamine exposure, yet little work has examined the main dietary contributors, particularly among children. We evaluated associations of diet with urinary melamine and derivative concentrations among 123 children aged 4–6 years in the Global Alliance to Prevent Prematurity and Stillbirth cohort. Children’s diets on the day preceding urine collection were assessed using 24-h dietary recalls. Associations of meat, fruit, and grain intakes with melamine exposure were examined using multiple linear regression. Remaining food groups were examined in secondary analyses. Mean (SD) melamine, ammelide, and cyanuric acid concentrations were 6.1 (12.4), 1.9 (2.1), and 60.6 (221.2) ng/mL, respectively. The second tertile of red meat consumers had 98% (95% CI: 15%, 241%) greater melamine exposure than non-consumers, yet the highest consumers did not have increased exposure. Greater consumption of certain fruits was associated with lower urinary ammelide. The top yogurt consumers had 112% (95% CI: 29%, 247%) greater melamine exposure than non-consumers. Consumption of starchy vegetables excluding potatoes was associated with 139% (95% CI: 6%, 437%) greater urinary ammelide. These observed associations should be confirmed in future studies using larger samples and increased monitoring of non-dietary routes of exposure.

## 1. Introduction

Melamine is a high-production-volume chemical with many industrial uses, including the manufacture of laminates, dinnerware, flooring, adhesives, insulation foams, and cleaning products. In high doses, ingestion of melamine can induce urinary stone formation by forming a complex with a related compound, cyanuric acid, or with endogenous uric acid [1]. Crystal formation within nephrons leads to progressive tubular blockage and degeneration, potentially resulting in kidney injury and death [2]. Studies from the US and other countries have shown that low-level exposure to melamine is ongoing [3,4] and can contribute to kidney stone formation and injury [5,6], particularly in the setting of other risk factors, such as small body size, chronic kidney disease, low fluid intake, or co-ingestion of cyanuric acid [2].

Diet is thought to be a primary pathway of melamine exposure. Melamine is present in food packaging materials [7] and can migrate into foods from melamine-containing tableware or utensils throughout the life of those items [8,9]. Furthermore, contamination of water and soil may result in melamine uptake by crops destined for the food supply [10]. Use of the pesticide and veterinary drug, cyromazine, which is metabolized to melamine, may also contribute to human exposure through ingestion of affected animal products [11]. Melamine has been detected in contaminated foods along with ammeline and ammelide, two derivatives of melamine that also arise as impurities of melamine production [7]. Cyanuric acid is another structural analogue that can arise from microbial degradation of melamine, but also has unique routes of exposure among humans, as it is commonly used in the disinfectant dichloroisocyanurate [12].

Although dietary pathways of melamine exposure have been well documented [7,9,13], relatively little work has examined the primary foods or dietary behaviors associated with melamine exposure in humans. Using data from the National Health and Nutrition Examination Survey (NHANES), we previously found that processed meats, whole grains, and possibly other plant-based foods may be associated with greater melamine exposure [14]. Data from Shanghai suggested that rice, fruits, meat, and eggs may be key sources of melamine exposure [4]. Direct measurements of foods sampled in Albany, NY showed that melamine concentrations were greatest in dairy products, cereal products, and meats [7]. Importantly, these reports give little insight into the dietary sources of melamine exposure among children, whose dietary intake, tableware usage, and other lifestyle factors may differ from adults. Further study is warranted in this vulnerable age group, as melamine may cause lasting kidney damage among children with the greatest exposure [15] and may additionally disrupt neurodevelopment or endocrine function [16].

The purpose of this study was to examine exposure to melamine and related compounds among children aged 4 to 6 years in the PATHWAYS GAPPS (Global Alliance to Prevent Prematurity and Stillbirth) cohort and to investigate potential dietary sources of exposure. We hypothesized that consumption of meats, fruits, and grain products would be positively associated with exposure to melamine. We also expected that greater consumption of foods prepared outside the home (versus at home) may be associated with greater exposure due to increased contact with materials that could contaminate the finished products.

## 2. Methods

### 2.1. Participants

This study used data collected from participants in the PATHWAYS GAPPS cohort. The GAPPS study enrolled pregnant women into a biorepository to aid research in preterm birth. Eligible women were 18 years or older, were able to speak and write in English, and were located in Seattle, WA or Yakima, WA. Women previously enrolled in GAPPS at these two sites were re-contacted and invited to participate in PATHWAYS, an NIH-funded study within the ECHO (Environmental influences on Child Health Outcomes) consortium if their child was born between 2011 and 2015. The current study included children who participated in the follow-up at age 4–6 years. At this study visit, sociodemographic and household data were gathered from maternal questionnaires and spot urine samples were collected.

### 2.2. Dietary Assessment

Participants reported their dietary intake on the day preceding the study visit with the assistance of their parent using the Automated Self-Administered 24-h (ASA24^®^) Dietary Assessment Tool, a web-based platform that uses the Automated Multiple Pass Method to collect information on the foods and beverages consumed on a given day [17]. Children’s consumption of various food groups was estimated using the USDA Food Patterns Equivalents Database (FPED) 2015–2016. The FPED converts reported foods and beverages into the number of cup equivalents of fruit, vegetables, and dairy; ounce equivalents of grains and protein foods; teaspoon equivalents of added sugars; and gram equivalents of solid fats and oils. Based on previous studies of dietary melamine sources [4,7,14], the primary analysis in this study included 4 variables for fruit (total fruit, citrus/melons/berries, fruit juice, and other fruit), 3 variables for grains (total grains, whole grains, and refined grains), and 8 variables for protein-rich foods (total protein foods, total meats, processed meats, red meats excluding processed meats, poultry, seafood rich in n-3 fatty acids, seafood low in n-3 fatty acids, and eggs). The secondary analysis included vegetables, dairy, oils, solid fats, and added sugars.

### 2.3. Classification of Food Source

An exploratory analysis was conducted to evaluate the associations between food sources and melamine exposure. We categorized participants into those whose foods came primarily from outside the home (“food away from home”, [FAFH]) versus those whose foods were prepared inside the home (“food at home”, [FAH]), as this distinction may be an important determinant of packaging and handling procedures that could introduce melamine contamination. Participants reported the source of each food item in the ASA24^®^ from an existing list of choices or by entering a unique description. Each source was categorized as either FAH of FAFH based on definitions outlined by the USDA Economic Research Service [18]. In general, foods prepared within the home and obtained from retail establishments like grocery stores and supercenters are considered FAH. FAFH is obtained from sources such as restaurants, convenience stores, vending machines, and school cafeterias. Each participant’s percentages of total energy intake from FAH and FAFH were calculated. The top half of the distribution of FAH consumers, who consumed ≥80% of total energy from FAH, were classified as high FAH consumers. Participants who consumed less than this threshold of FAH were classified as high FAFH consumers.

### 2.4. Melamine Analysis

Spot urine samples were used to analyze concentrations of melamine and three related compounds: ammeline, ammelide, and cyanuric acid are analogous to melamine with one, two, and three amino groups replaced by hydroxyls, respectively. The extraction and analysis of these compounds in urine have been previously described in detail [19]. Briefly, two aliquots of urine were prepared. One was acidified with 1% formic acid for cyanuric acid analysis and the other was alkalized with 5% ammonium hydroxide for the analysis of melamine, ammeline, and ammelide. The samples were extracted twice, and the combined extract was concentrated to near-dryness under nitrogen, reconstituted, and filtered into glass vials prior to high performance liquid chromatography-tandem mass spectrometry (HPLC-MS/MS) analysis. Melamine, cyanuric acid, ammeline, and ammelide were analyzed using a Shimadzu LC-30 AD Series HPLC system (Shimadzu Corporation, Kyoto, Japan) connected to an API 5500 triple-quadrupole MS/MS (Applied Biosystems, Foster City, CA, USA). The limits of detection (LOD) for melamine, ammelide, cyanuric acid, and ammeline were 0.08, 0.15, 0.10 and 0.10 ng/mL, respectively.

### 2.5. Statistical Analysis

Mean and standard deviation (SD) of urinary melamine and derivative concentrations were summarized by sociodemographic subgroups within the study sample, and *t*-tests or ANOVA were used to evaluate differences in exposure across groups. Prior to conducting *t*-tests or ANOVA, urinary concentrations were log-transformed due to right skew.

Multiple linear regression modeling was used to examine potential dietary sources of exposure to melamine and ammelide. Although cyanuric acid can be found in dietary sources [7], it was not considered in these analyses because of its distinct industrial applications and routes of human exposure [12]. Ammeline was also not considered in these analyses due to lack of detection among the samples in this study. The dependent variable was melamine or ammelide urinary concentration adjusted for specific gravity using the formula proposed by Levine and Fahy [20] and then log-transformed for improved model fit. Concentrations of analytes that fell below the LOD were imputed as the LOD divided by the square root of two [21].

For foods consumed by ≥80% of participants, intakes were log transformed after adding 1 to accommodate the small number of zeros [22,23], and the transformed variable was used as a continuous term for modeling. Foods consumed by ≤20% of the participants were classified as binary variables (i.e., consumers versus non-consumers), as shown in Table A1. Foods consumed by at least 25 but fewer than 35 individuals were classified as non-consumers and low or high consumers, split at the midpoint of ranked consumers. For remaining foods, which were consumed by at least 35, participants were classified as non-consumers versus tertiles of consumers.

Model covariates were chosen a priori based on existing literature [14,19]. Minimal models adjusted for age and sex, while full models additionally adjusted for race and ethnicity (non-Hispanic White, Hispanic White, multiple races, or other), maternal education (less than a college degree completed, college graduate, or advanced degree), household income category ($45,001 to 75,000; 75,001 to 100,000; 100,001 to 150,000; or 150,001 or greater), BMI percentile, study site (Seattle, Yakima), and pesticide use in the home during the past 12 months (yes, no). Of the 123 participants with dietary and melamine data, one was missing BMI, and one was missing household income data. Thus, minimally adjusted models included 123 participants and full models included 121. Model fit was assessed through visual inspection of residuals for homoscedasticity and normality. Due to variable transformations, regression coefficients were back-transformed to derive the expected percentage difference in specific gravity-adjusted analyte concentrations in comparison to the reference group for categorical dietary variables, and to derive the expected percentage difference in analyte concentrations for a 50% increase in continuous dietary variables. The level of statistical significance was set at *p* < 0.05.

## 3. Results

Urinary melamine concentrations were above the LOD in all participants’ urines, with a mean (SD) of 6.1 (12.4) ng/mL (Table 1). Concentrations of ammelide and cyanuric acid were above the LOD in 99.2% and 87.8% of participants, respectively. Mean (SD) ammelide concentrations were 1.9 (2.1) ng/mL and mean cyanuric acid concentrations were 60.6 (221.2) ng/mL. Two participants had cyanuric acid concentrations greater than 1000 ng/mL, contributing to the large SD. Ammeline concentrations were below the LOD of 0.1 ng/mL in all samples (not shown).

Melamine concentrations were higher among males compared to female participants (*p* = 0.04), but ammelide and cyanuric acid concentrations were similar between sexes (Table 2). Exposure to melamine and cyanuric acid appeared to be greater in Seattle compared to Yakima, yet these did not meet the threshold for statistical significance (*p* = 0.12 and 0.13, respectively). There were no significant associations between analyte concentrations and maternal education or income, although higher educational attainment appeared to be associated with greater cyanuric acid concentrations (*p* = 0.15). Children whose BMI percentiles were in the obese range tended to have lower urinary concentrations of each analyte, although not statistically significantly. Among the 15 participants who reported pesticide use in the home over the last year, mean (SD) urinary melamine was 8.1 (7.5) ng/mL in comparison to 5.8 (13.0) ng/mL among those who hadn’t used pesticides (*p* = 0.05). Children who predominantly consumed FAH tended to have lower urinary concentrations of each analyte than those who consumed more FAFH, although not statistically significantly (*p* = 0.84, 0.13, and 0.67 for melamine, ammelide, and cyanuric acid, respectively).

Because results of the minimally and fully adjusted models were similar (Table A1), we present results of full models only in Figure 1 and Figure 2. The second tertile of red meat consumers had 98% (95% confidence interval, CI: 15%, 241%) greater melamine exposure than non-consumers (Figure 1). However, this association did not exhibit a clear dose–response relationship, as the greatest red meat consumers did not have increased exposure compared to non-consumers. The highest consumers of citrus, melons, and berries had 56% lower (95% CI: −79%, −11%) urinary ammelide concentrations than non-consumers. No other category of fruit, grain, or meat consumption was associated with differences in ammelide or melamine exposures.

In the secondary analysis, the top yogurt consumers had 112% (95% CI: 29%, 247%) greater melamine exposure than non-consumers (Figure 2). Consumption of starchy vegetables excluding potatoes was associated with 139% (95% CI: 6%, 437%) greater exposure to ammelide. No relationships of dietary fats or added sugars with ammelide or melamine were observed.

The exploratory analysis generally showed trends towards greater exposure to melamine and ammelide with greater consumption of FAFH, but associations were not statistically significant in either minimally or fully adjusted models (Table 3).

## 4. Discussion

We observed ubiquitous exposure to melamine and related compounds among children in the GAPPS cohort and identified red meat, certain starchy vegetables, and yogurt as potential dietary contributors. Children in this analysis had median urinary melamine concentrations of 3.0 ng/mL, which was similar to the median value of 4.7 ng/mL observed among 109 children aged 4 months to 8 years in other cohorts from Seattle, WA and New York City, NY [6]. We previously observed a mean melamine exposure of 0.7 ng/mL among US children under 18 years of age in NHANES 2003–2004 [14], which was somewhat lower than the mean of 6.1 ng/mL in this study. Taken together, these findings suggest pervasive ambient exposure to melamine and its derivatives among US children.

A previous analysis of 35 children younger than 3 years suggested that urinary melamine concentrations greater than 7.1 µg/mmol creatinine represent significant exposure to melamine-tainted products [24]. The mean of 1.0 (SD: 1.7) µg/mmol creatinine observed in this study was well below this threshold, although melamine exposures of two participants (1.6%) exceeded 7.1 µg/mmol creatinine, indicating possible risk for melamine-related stones [24]. Importantly, exposures observed in this study could be associated with earlier signs of kidney injury. Analysis of other childhood cohorts with similar mean cyanuric acid exposure (35.3 ng/mL) to the present study (60.6 ng/mL) found that cyanuric acid exposure was associated with early kidney injury, as indicated by kidney injury molecule 1 [6].

This analysis showed that two animal-based food sources may contribute to melamine exposure: red meat and yogurt. Although we did not observe a clear dose-dependent association of red meat intake with melamine exposure, our finding of increased melamine exposure among some higher consumers was consistent with previous reports. Shi et al. noted positive associations of beef and mutton consumption with melamine exposure among adults in Shanghai [4], and we previously observed an association of processed meats with melamine exposure in NHANES [14]. Meats and dairy products stood out among the most highly contaminated foods analyzed in New York state [7]. In the same analysis, dairy products were estimated to be the single largest contributor to total dietary melamine exposure among children aged 1–6 years [7], which is consistent with our finding regarding yogurt’s association with melamine exposure.

There are multiple routes by which animal-based food products may become contaminated with melamine or related compounds. Cyanuric acid is authorized for use in the US as a component in feed-grade biuret, a source of nonprotein nitrogen in animal feed [25]. Cyanuric acid and melamine have been found ubiquitously in bovine feed from China, India, and the US [26]. Several studies have documented that ingested melamine can subsequently be detected in the tissues of pigs [27], sheep [28], and chickens [29,30]. Melamine in animal feed can also be transferred to the milk and tissues of dairy cows [31] and the eggs of laying hens [32]. Another source of melamine exposure for agricultural animals is the pesticide cyromazine. Cyromazine is used as a feed-through fly control agent for horse and chicken manure, and can be metabolized in these animals to melamine [11,12]. Thus, it is unsurprising that the present study and others [4,7] have identified multiple animal-based foods as key sources of human exposure.

While most fruits, vegetables, and grain products were not associated with exposure to melamine or ammelide in this study, certain starchy vegetables were positively associated with urinary ammelide. This may be explained by cyromazine use on agricultural crops. Like animals, plants may metabolize cyromazine to melamine or related metabolites [33]. Among several other vegetables, cyromazine is commonly used on root crops [33], many of which would be classified into the starchy vegetable category in this analysis. It was unexpected to observe a negative association between some fruits and ammelide exposure, and we cannot rule out spurious correlation. Alternatively, higher consumption of these fruits may have displaced other foods in children’s diets which may be more prone to melamine contamination.

We were also somewhat surprised to find no association between food source (i.e., FAH versus FAFH) and melamine exposure in this study. FAFH is often associated with increased use of food packaging materials, which have been noted to be potential melamine sources [7]. However, many FAH sources could have significant contact with similar food packaging materials, especially as supermarkets shift their offerings to provide more convenient packaged foods. Melamine may also be used in food contact materials that are not unique to FAFH. For example, melamine–formaldehyde resins, which are often used in can coatings and the metal lids of glass jars, can release melamine into foods [34].

One important limitation of this study is that we could not account for participants’ potential use of melamine tableware. Migration from melamine tableware has been well-documented in controlled studies [8,9,35], especially with acidic foods, high temperatures, and extended contact times [36]. One small intervention study demonstrated that switching from melamine-containing tableware and utensils to stainless steel alternatives reduced melamine exposure by approximately 68% [37]. Therefore, among participants who may have used melamine tableware, associations of specific foods with exposure levels would have been challenging to identify in this study. We were also unable to differentiate between organic or conventionally grown foods in participants’ diets, which could influence exposure to melamine or related compounds because of cyromazine usage. The use of a single spot urine sample to assess melamine exposure is another limitation of this study. Although melamine concentrations of spot urine samples have been found to be highly correlated with excretion across several days in children [38] and adults [19], diurnal variations may exist due to the short half-life of melamine [12]. Thus, spot urine samples may not reflect exposure during the entirety of the dietary data collection period. Additionally, this analysis was not able to account for numerous non-dietary sources of exposure to melamine. For example, dust is one principal source of exposure [39,40]. Melamine and related compounds have also been found in mats used for children’s naps, which could lead to significant dermal absorption [40]. Children may be exposed to melamine or its derivatives through other household items such as flooring materials, cleaning products, and paint pigments [41]. Textiles used in children’s clothing, especially before their initial washing, may be another source of exposure [42].

## 5. Conclusions

This study is the first to our knowledge to evaluate associations of dietary intake with urinary melamine and derivative concentrations in a childhood cohort. These data demonstrate ubiquitous exposure of children in the GAPPS cohort to melamine and related compounds, and identified red meat, certain starchy vegetables, and yogurt as potential dietary sources of exposure. Studies with larger sample sizes and increased monitoring of non-dietary sources of exposure should be conducted to confirm these findings. Given that even low-dose melamine exposures may promote kidney damage in healthy individuals [43], additional study on the routes of exposure to melamine and related compounds is warranted.

## Figures and Tables

**Figure 1 ijerph-19-04964-f001:**
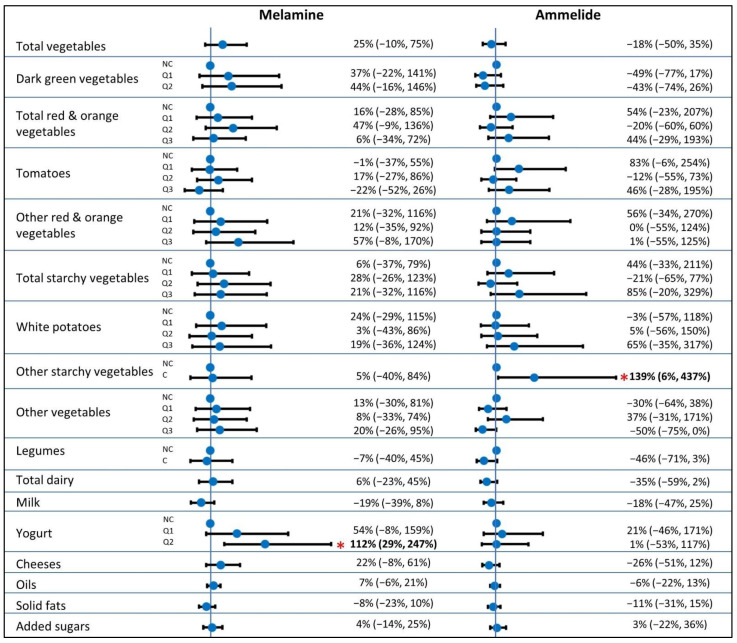
Estimated percentage difference (with 95% CI) in urinary melamine or ammelide concentrations for a 50% increase in consumption (continuous dietary variables) or in comparison to the reference group (categorical dietary variables) for fruits, grains, and protein foods. Estimates with *p*-values < 0.05 shown in bold text with asterisk. Models adjusted for age, sex, child race and ethnicity (Non-Hispanic White, Hispanic White, multiple races, other), household income (<45 k, 45–75 k, 75–100 k, 100–150 k, >150 k), maternal education (<college grad, college grad, advanced degree), BMI for age percentile (continuous), pesticide use in home over the last year (yes, no), and site (Seattle, Yakima); Abbreviations: NC, non-consumer; C, consumer; Q1–Q3, quantiles 1–3.

**Figure 2 ijerph-19-04964-f002:**
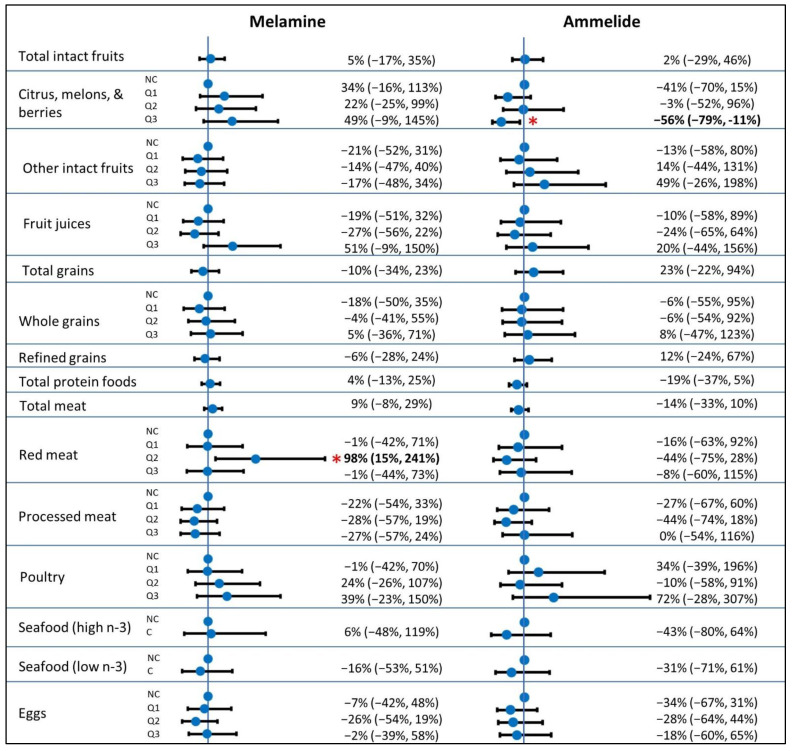
Estimated percentage difference (with 95% CI) in urinary melamine or ammelide concentrations for a 50% increase in consumption (continuous dietary variables) or in comparison to the reference group (categorical dietary variables) for vegetables, dairy, fats, and sugars. Estimates with *p*-values < 0.05 shown in bold text with asterisk. Models adjusted for age, sex, child race and ethnicity (Non-Hispanic White, Hispanic White, multiple races, other), household income (<45 k, 45–75 k, 75–100 k, 100–150 k, >150 k), maternal education (<college grad, college grad, advanced degree), BMI for age percentile (continuous), pesticide use in home over the last year (yes, no), and site (Seattle, Yakima); Abbreviations: NC, non-consumer; C, consumer; Q1–Q3, quantiles 1–3.

**Table 1 ijerph-19-04964-t001:** Distribution of melamine and related compounds in spot urine samples of 123 children aged 4–6 years in the PATHWAYS GAPPS cohort.

	Melamine (ng/mL)	Ammelide (ng/mL)	Cyanuric Acid (ng/mL)
Number (%) > LOD ^1^	123 (100%)	122 (99.2%)	108 (87.8%)
Mean (SD)	6.1 (12.4)	1.9 (2.1)	60.6 (221.2)
Minimum	0.41	0.02	0.02
Maximum	104.3	25.8	2110.0

^1^ LOD for melamine, ammelide, and cyanuric acid were 0.08, 0.15, and 0.10 ng/mL, respectively; ammelide and cyanuric acid values below LOD imputed as LOD divided by the square root of 2. Abbreviations: LOD, limit of detection; SD, standard deviation.

**Table 2 ijerph-19-04964-t002:** Concentrations of melamine, ammelide, and cyanuric acid (ng/mL) in spot urine samples among children aged 4–6 in the PATHWAYS GAPPS cohort and by sociodemographic subgroup.

		Melamine		Ammelide		Cyanuric Acid	
	n	Mean (SD)	*p* ^1^	Mean (SD)	*p*	Mean (SD)	*p*
**Total**	123	6.1 (12.4)		1.9 (2.1)		60.6 (221.2)	
**Sex**			**0.04**		0.71		0.54
Female	64	4.8 (6.7)		1.6 (1.1)		34.9 (55.0)	
Male	59	7.5 (16.5)		2.2 (2.9)		88.4 (313.2)	
**Site**			0.12		0.35		0.13
Seattle	45	7.9 (17.9)		1.6 (1.1)		95.9 (358.4)	
Yakima	78	5.1 (7.7)		2.1 (2.5)		40.2 (54.4)	
**Maternal Education**			0.34		0.83		0.15
<College graduate	51	4.6 (7.1)		2.0 (2.9)		41.0 (65.6)	
College graduate	42	7.7 (18.6)		1.8 (1.3)		54.7 (188.7)	
Advanced degree	30	6.4 (8.4)		1.7 (1.4)		102.2 (381.5)	
**Annual Income ^2^**			0.68		0.76		0.64
<$45,000	23	5.5 (9.4)		1.7 (1.4)		45.3 (79.3)	
$45,000 to 75,000	22	9.0 (25.3)		1.7 (1.1)		16.2 (12.9)	
$75,000 to 100,000	23	7.0 (6.6)		1.7 (1.5)		32.3 (30.0)	
$100,000 to 150,000	28	5.1 (7.8)		2.4 (3.7)		43.2 (64.1)	
>$150,000	26	4.7 (4.8)		1.8 (1.4)		157.4 (463.7)	
**Child Race and Ethnicity**			0.69		0.56		0.70
Non-Hispanic White	69	4.7 (6.3)		2.1 (2.7)		72.8 (256.3)	
Hispanic White	21	11.0 (26.8)		1.8 (1.3)		24.0 (21.5)	
Multiple races	17	5.6 (5.5)		1.4 (1.2)		101.2 (294.5)	
Other	16	6.2 (7.0)		1.5 (0.8)		12.7 (13.1)	
**BMI Status ^2^**			0.57		0.56		0.42
Underweight	1	2.3		3.4		22.0	
Healthy Weight	77	7.3 (15.2)		2.1 (2.6)		70.0 (243.8)	
Overweight	21	4.4 (4.5)		1.4 (0.9)		75.0 (263.4)	
Obese	23	4.2 (5.3)		1.6 (1.3)		20.2 (23.7)	
**Pesticide use in home ^3^**			0.05		0.45		0.75
No	108	5.8 (13.0)		1.9 (2.3)		54.6 (207.1)	
Yes	15	8.1 (7.5)		1.9 (0.9)		103.8 (310.3)	
**Food Source Category ^4^**			0.84		0.13		0.67
FAH	62	5.1 (1.7)		1.7 (1.5)		49.3 (158.9)	
FAFH	61	7.1 (16.8)		2.1 (2.7)		72.0 (271.2)	

^1^*p*-values generated from *t*-test or ANOVA; concentrations were log-transformed prior to testing due to right skew; *p*-values < 0.05 in bold text. ^2^ Household income and BMI data were each missing for one participant. ^3^ Pesticide use in the home was based on self-reported use over the past year. ^4^ Participants were classified as FAH consumers if ≥80% of total energy was derived from FAH sources or as FAFH consumers if <80% of total energy was derived from FAH. Abbreviations: SD, standard deviation; BMI, body mass index; FAH, food at home; FAFH, food away from home.

**Table 3 ijerph-19-04964-t003:** Estimated regression coefficients (95% CI) for associations of FAFH (versus FAH) with urinary melamine and ammelide concentrations.

	Minimal ^1^	Full ^2^
Melamine	0.04 (−0.29, 0.36)	0.15 (−0.20, 0.49)
Ammelide	0.36 (−0.10, 0.81)	0.27 (−0.25, 0.78)

^1^ Minimal models adjusted for age and sex. ^2^ Full models additionally adjusted for child race and ethnicity (Non-Hispanic White, Hispanic White, multiple races, other), household income (<45 k, 45–75 k, 75–100 k, 100–150 k, >150 k), maternal education (<college grad, college grad, advanced degree), BMI for age percentile (continuous), pesticide use in home over last year (yes, no), and site (Seattle, Yakima).

## Data Availability

Not applicable.

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
