# Peer review of "Associations of Dietary Intake with Urinary Melamine and Derivative Concentrations among Children in the GAPPS Cohort"

_ijerph, 2022, doi:10.3390/ijerph19094964_

Round 1

Reviewer 1 Report

The work deals with a very interesting topic. However, I would like to ask you to clarify the characteristics of the database and the methods of how it was used and calculated at work. This type of information is missing from the description of the methods.

Author Response

Thank you for reviewing our manuscript and providing your feedback. We added a graphical abstract to our manuscript that we hope will help readers quickly gather key information about the study question, population, methods, and findings. More detailed information is found in the Methods section.

Section 2.1 (Methods – Participants) describes the history of the cohort from which data were gathered and analyzed for this study. We would not consider this as a “database” study, as referred to in your comment. To clear up any confusion about the source of our data, we added the sentence “This study used data collected from participants in the PATHWAYS GAPPS cohort” at the beginning of the Methods section.

Our Methods section also gives specific details about the collection and processing of dietary data in Sections 2.2 (Dietary Assessment) and 2.3 (Classification of Food Source). We provide details on the methods used to assess urinary concentrations of melamine and its derivatives in Section 2.4 (Melamine Analysis). Finally, Section 2.5 (Statistical Analysis) provides a detailed summary of the steps used to analyze data.

Reviewer 2 Report

Manuscript ID: ijerph-1641449

Title: Associations of Dietary Intake with Urinary Melamine and Derivative Concentrations among Children in the GAPPS Cohort

The manuscript is scientifically sound and is well written for publication in the International Journal of Environmental Research and Public Health. The main subject of the proposed text is about dietary intake of melamine and derivates among children. The emphasis of the proposed manuscript is placed on urinary melamine and derivate concentrations among children. I would recommend the paper for publication after some clarification listed below:

Lines 178-186 Could not find mean (SD) values in Table1. The SD value of cyanuric acid is high, please could you check it.

Table 2. SD values of cyanuric acid are high. Could you check it?

Best regards!

Reviewer 3 Report

Aim of this study was to evaluate the association of diet with urinary melamine and related compounds among 123 children aged 4 to 6 years and to investigate potential dietary sources of exposure. Starting hypothesis was that the consumption of meats, fruits, and grain products would be positively associated with exposure to melamine. Moreover, the consumption of foods prepared outside the home (FAHF) with respect to home (FAH) could be associated with greater exposures.  The concentration of melamine and derivates (ammelide, cyanuric acid) was measured by collecting spot urine samples from the participants children. The results indicate urinary melamine above LOD in all participants’ urines, while ammelide and cyanuric acid above LOD in a large part of them and  ammeline below in all samples.

Despite the interest of the subject of the study, even more because it is conducted on children - the negative consequence of these compounds intake are well known and here described- the manuscript is confusing and unreadable so that in the present version it is far to be acceptable for publication.

Briefly, the authors in analyzing the dietary routes exposure to melamine and derivatives identify three possible causes: 1) packaging materials, 2) the use of cyromazine on agricultural crops that can be metabolized by plants and vegetables as well as animals, and 3) the use of melamine-containing tableware and utensils instead of stainless-steel alternatives capable of reducing melamine exposure. At the same time, they declare to be surprise to find no association between  foods prepared FAFH with respect FAH as the first could be associated with greater exposures due to an increased contact with materials that could contaminate the products.

In this reviewer’s opinion, the authors consider too overlapping possible causes, none of them further investigated. Therefore I suggest that the study must be completed and the manuscript improved as follows:

  • The authors have to well clarify at which concentration these substances would be dangerous to the subjects’ health in comparison to the levels found in the study;
  • Additional experiments must be carried out to investigate the incidence of each of the items taken into consideration.
  1. Packaging materials: were all packaging considered in the study made of the same materials? The authors have to clarify this point. Otherwise, which are the materials they consider more dangerous under this respect and the results were in accordance with their hypothesis? Then the authors could take in consideration packaging materials not plastic, but paper-made and investigate by new experiments a reduction, possible below LOD, of the compounds’ concentration.
  2. The authors take in consideration a large part of foods: vegetables, fruit, meat, yogurt and in a summary fashion present and discuss the difference in the exposure and the possible explanation of the data. Additional experimental part would be requested to compare their results with data obtained from food coming from organic cultivation.
  3. The incidence of the use of melamine-containing tableware on the data must be matter of investigation and this could be made by comparing data obtained from the use of these table ware versus the other type of utensils.

In this reviewer’s opinion each of these points plays great interest, but it was too complicated to take in consideration all of them at the same time. Each of them, properly investigated as explained above, could be, alone, matter of investigation and publication. The reader of this paper becomes aware that all steps in the dietary route is a sure source of contamination but he doesn’t receive  information about how bad it is for health nor how it is possible to avoid or at least reduce this dangerous situation.

  • Finally, as told above, the manuscript is unreadable in the present form mainly due the absence of graphical elements. The very big tables are not well explained and very boring to analyze. They must be substituted by histograms and/or graphs and /or any graphical element to make the reader easily capable of understanding the scientific message.

Round 2

Reviewer 3 Report

In the light of the modifications and the

answers to the raised criticisms, 
this reviewer think the manuscript

suitable to be published.